# Applying the open-LUCIS framework to identify and characterize human–wildlife conflicts: A case study in Botswana

Silas Achidago[1]*, Changjie Chen[2], Jasmeet Judge[3], Mogae Makonyela[4],
Lynn Fanikiso[4], Lara Sousa[5], Robynne Kotze[4], Gregory Kiker[3], Kedisaletse Selume[4],
Kim Young[6], Robin Lines[7], Jess Isden[4], Andrew Loveridge[4,5,6], Yan Wang[2], Aditya Singh[3]

**1** University of Florida, Department of Urban and Regional Planning, Gainesville, Florida, United States of America, **2** Florida Institute for Built Environment Resilience, University of Florida, Gainesville, Florida, United States of America, **3** Center for Remote Sensing, Institute of Food and Agricultural Sciences, Department of Agricultural and Biological Engineering, University of Florida, Gainesville, Florida, United States of America, **4** WildCAT Botswana Trust, Maun, Botswana, **5** WildCRU, Department of Biology, University of Oxford, Oxford, United Kingdom, **6** Panthera, New York, New York, United States of America, **7** Durrell Institute of Conservation and Ecology, University of Kent, Canterbury, United Kingdom

* achidago@gmail.com

## Abstract

Human–Wildlife Conflict (HWC) is an increasing challenge in rapidly changing landscapes, where agricultural expansion, settlement growth, and infrastructure development intersect with critical wildlife corridors. Addressing these conflicts requires spatially explicit methods that can evaluate trade-offs among competing land uses. This study demonstrates the application of the open-source Land Use Conflict Identification Strategy (Open-LUCIS), a suitability-based framework that integrates open geospatial data, domain knowledge, and goal-driven land-use modeling. Using Pandamatenga in Botswana's Chobe District as a case study, we identified areas of potential conflict among agriculture, human settlement, and wildlife conservation. High-conflict zones were concentrated where commercial farms overlap with transboundary wildlife corridors, highlighting the tension between agricultural development and conservation. A sensitivity analysis indicated that existing land use, road accessibility, and development constraints strongly influence conflict dynamics. The application demonstrates a clear pathway for using open-source tools to support HWC studies. By relying on open data and reproducible methods, Open-LUCIS offers a cost-effective and accessible alternative to proprietary software, with direct implications for advancing sustainable land development in regions with limited resources. Given that the dynamics observed in Chobe reflect pressures common across many parts of Africa and beyond, the framework is broadly applicable as a transferable approach for managing land-use conflicts in many rapidly developing, ecologically sensitive frontiers worldwide.

**Data availability statement:** All data sources except the wildlife habitat suitability, wildlife-related data, farm and settlement layout data used in the study are available for download as referenced in Table 1—Data Summary with URLs to the download pages. These data cannot be shared publicly because of a data use agreement signed with WildCAT Botswana Trust, who provided the data. Data are available from WildCAT, Botswana Trust for researchers who meet the criteria for access to confidential data. Contact person for WILDCAT Botswana Trust. Dr. Jess Isden Coexistence Coordinator +267 7577 5316 jess@wildcatbotswana.org; jesstkpp@gmail.com WildCAT Botswana Trust Office 13, Plot 448, Mathiba Road, Maun, Botswana WildCRU, University of Oxford, UK.

**Funding:** This study was supported by the NASA/USAID SERVIR Program (https://servir.icrisat.org/) under Grant Number 80NSSC20K0153 awarded to JJ and Grant Number 80NSSC23K0247 awarded to AS. Additional support was provided through Grant Number NA12481 from the Hamer Foundation and the World Wildlife Fund for Nature (https://hamerfoundation.org/) awarded to JI. JJ contributed to the study design and the decision to publish. AS and JI were involved in the decision to publish. The funders had no role in study design, data collection and analysis, decision to publish, or preparation of the manuscript.

**Competing interests:** The authors have declared that no competing interests exist.

## 1. Introduction

In many parts of the developing world, rapid population growth and expanding development are intensifying land-use pressures, fragmenting ecosystems, and heightening the risks of Human–Wildlife Conflict (HWC) [10–13].This is reflected worldwide in rising crop damage, livestock predation, property loss, and fatalities; in Kenya, HWC compensation reached $4.2 million in 2021–22, while in India, conflicts caused over one million hectares of crop loss and hundreds of human and elephant deaths [14,15]. Similar dynamics are unfolding in Botswana's Chobe District, but with sharper intensity given its exceptionally high elephant population and rapid agricultural expansion along critical transboundary corridors [16–18].

Chobe's HWC issue underscores the importance of effective land management, a challenge mirrored across the Global South and exemplified locally by institutions such as the Chobe Land Board (CLB). As the authority responsible for allocating and regulating land use in the district, the CLB faces the dual mandate of supporting agricultural development while safeguarding biodiversity corridors [19]. Addressing this challenge requires decision-support tools that can integrate ecological realities with human demands, providing planners with transparent and reproducible evidence. Conservation planning tools such as Marxan, InVEST, and Maxent are widely used to optimize protected areas, preserve biodiversity, and assess ecosystem quality [20,21]. While powerful, they primarily emphasize conservation or valuation objectives rather than balancing directly competing land uses. The Land Use Conflict Identification Strategy (LUCIS) framework takes a different approach: it is explicitly designed to identify and address conflicts among competing land demands through a GIS-based, multi-criteria process that integrates physical, economic, and social factors while incorporating stakeholder input [22,23]. To make this framework more transparent and accessible, Chen et al. [24] developed the PyLUSAT-QGIS plugin that operationalizes land suitability analysis in open-source platforms. This advancement underpins the development of Open-LUCIS, providing analysts and policymakers in resource-limited regions with a flexible, reproducible, and cost-effective method for HWC assessment.

In this study, we apply the Open-LUCIS framework to the Chobe District, using Pandamatenga—a designated Special Economic Zone and Botswana's agricultural hub [25]—as a focal case. The area is experiencing rapid settlement and agricultural expansion that directly intersects with critical wildlife corridors [19], making it an ideal setting to examine HWC. Specifically, we (a) conduct a land suitability analysis for agriculture, human settlement, and wildlife; (b) identify and map regions most vulnerable to HWC; and (c) assess the influence of key environmental and human factors through a sensitivity analysis. This case demonstrates how Open-LUCIS can support local institutions, such as the CLB, by providing transparent, reproducible evidence for land-use decisions.

## 2. Materials and methods

To establish a baseline scenario, we combined various geospatial datasets with domain knowledge provided by the WildCAT Botswana Trust (hereafter referred to

as WildCAT), a locally registered NGO under the Wildlife Conservation Research Unit (WildCRU) of the University of Oxford's Biology Department. WildCAT works with government agencies, research institutions, and local communities, ensuring that both scientific evidence and lived experience inform the evaluation of HWC. This case study followed the procedure as defined in the LUCIS model to combine multiple land suitability criteria. Moreover, the weights used in this procedure, as presented later in this paper, were developed in consultation with WildCAT and further refined through unofficial dialogs with CLB's technical officers. While this approach captured domain expert opinions, it did not constitute a formal stakeholder survey, the preferred method by LUCIS. Future applications should incorporate structured participation (e.g., surveys, interviews, role-playing games) to ensure locally grounded perspectives complement expert inputs, consistent with the participatory approach in land use planning.

## 2.1. Study area

Pandamatenga, as shown in Fig 1, is located at latitude 18º 32' South and longitude 25º 38' East, covering an area of 6,842 km$^2$ in northern Botswana. Characterized by a semi-arid climate, the Chobe District experiences hot and moist

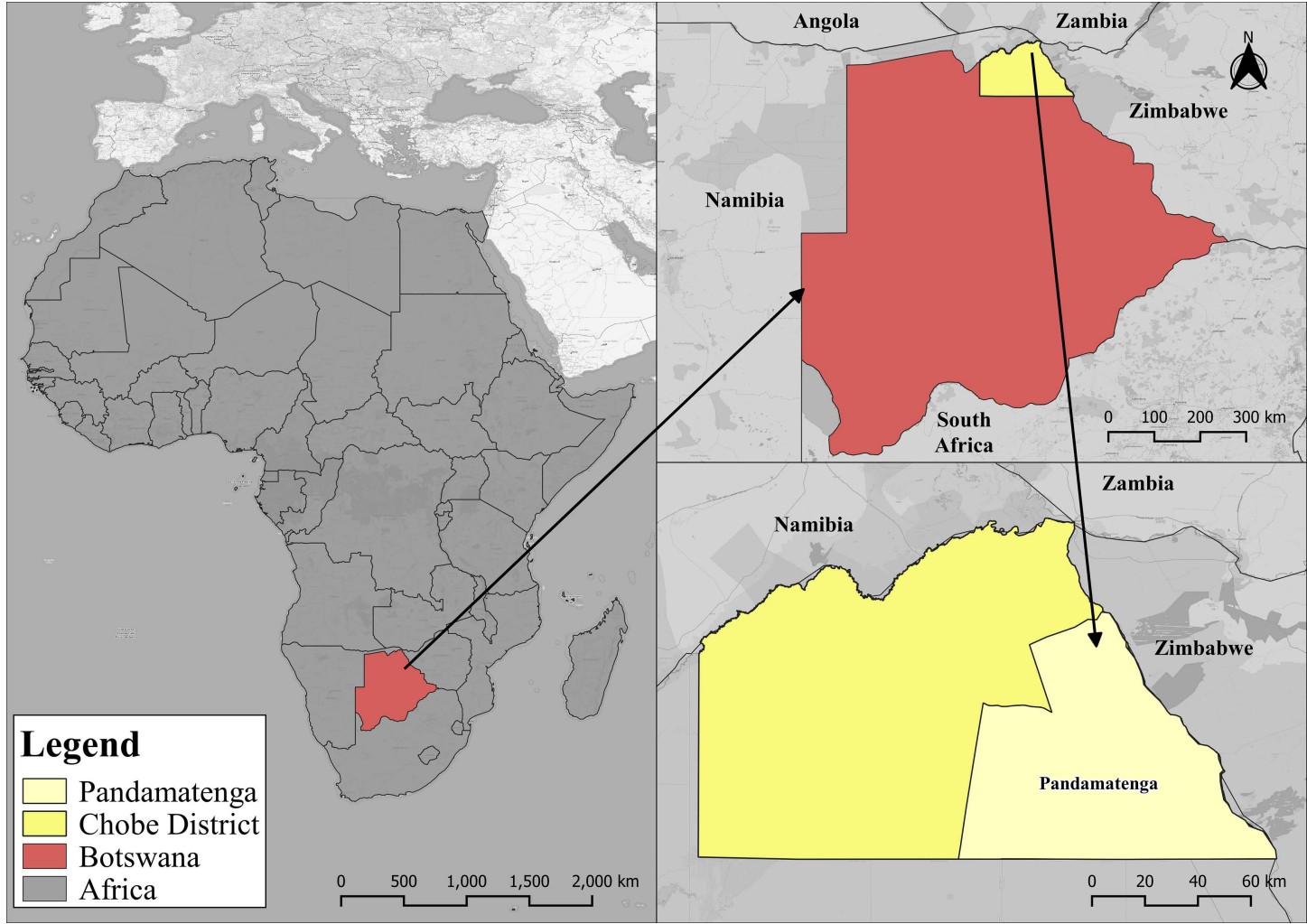

**Fig 1. Map showing the context and location of Pandamatenga.** Republished from OpenStreetMap Contributors under a CC BY license, with permission from OpenStreetMap Foundation, original copyright 2023.

summers, followed by dry to mild winters, with an average annual rainfall of around 640 mm, making it the wettest climate in Botswana [26].

Temperature variations in Pandamatenga are notable, with maximums ranging from 35°C to 40°C during the summer months of October to March, and minimums ranging from 11°C to 20°C between November and July [27]. The vegetation, as shown in Fig 2, is predominantly characterized by extensive grassland savannah, interspersed with mopane (*Colophospermum mopane*) and acacia species [28]. The area is generally flat with a gentle slope, and rainwater flows following natural drainage routes. Pandamatenga provides an important context for this study, as instances of human-wildlife conflicts have been frequently reported. Fig 3 illustrates the spatial distribution of these incidents across the area.

## 2.2. Data sources

Table 1 shows the summary of open-data GIS and remotely-sensed datasets provided by WildCAT. All datasets were harmonized to a common spatial resolution and/or projected to EPSG:32734 (WGS 84/UTM Zone 34S) to ensure spatial accuracy and consistency across analyses.

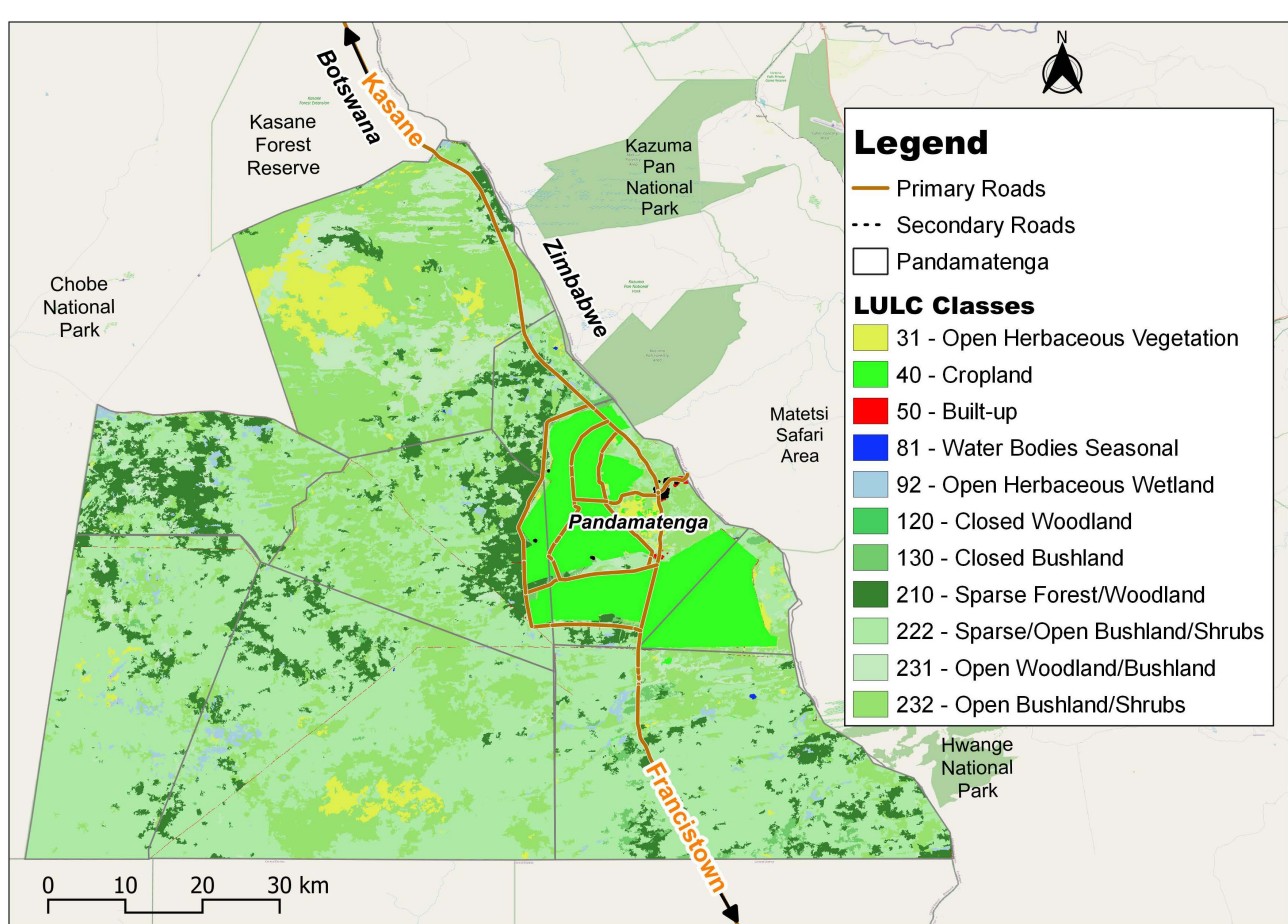

**Fig 2. Map showing Land Use Land Cover (LULC) of Pandamatenga.** Republished from OpenStreetMap Contributors under a CC BY license, with permission from OpenStreetMap Foundation, original copyright 2023.

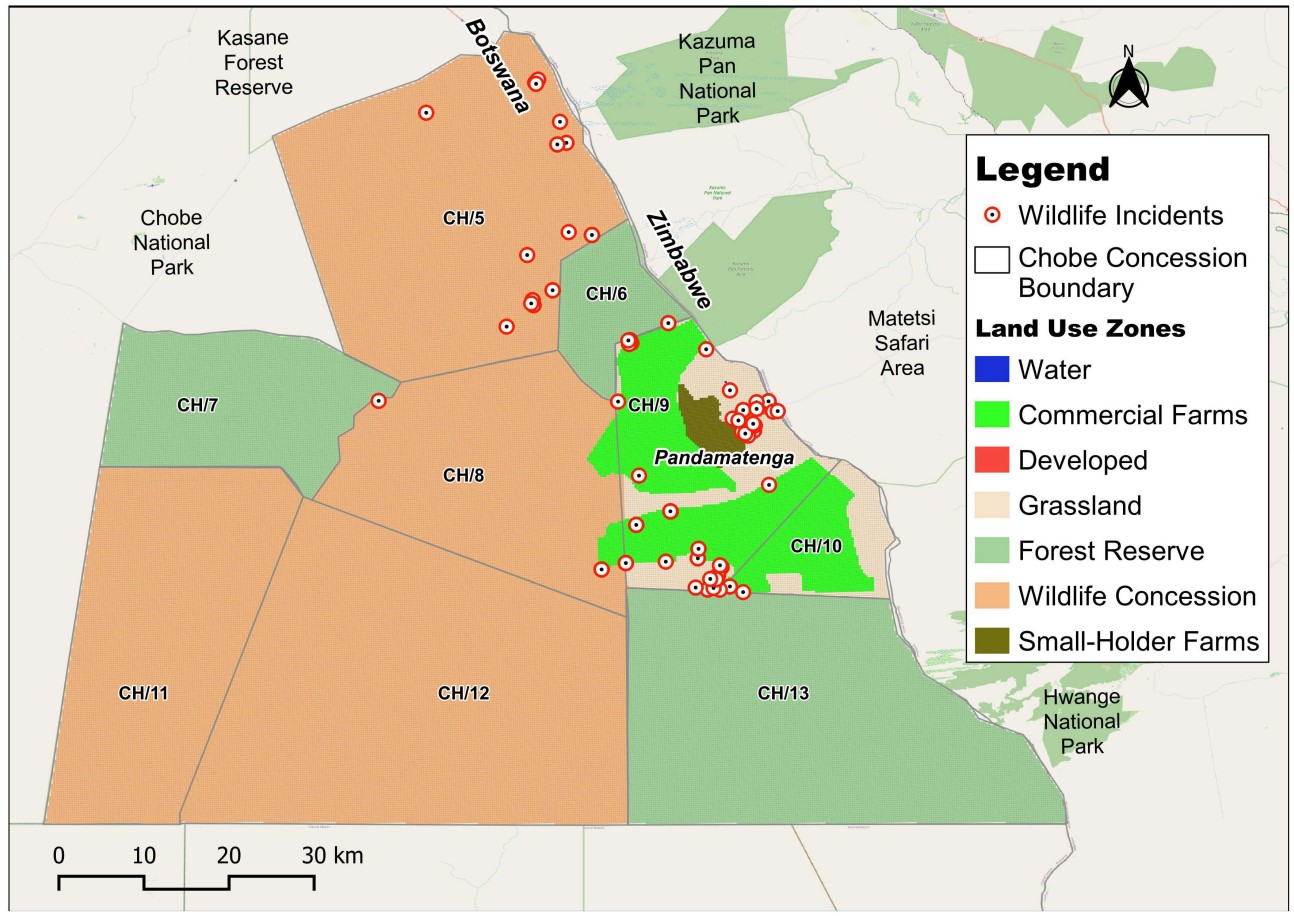

**Fig 3. Map showing existing human-wildlife incidents in Pandamatenga between 2007–2023 as reported by the Problem Animal Control – Dept.** Wildlife and National Parks, Botswana. Republished from OpenStreetMap Contributors under a CC BY license, with permission from OpenStreet-Map Foundation, original copyright 2023.

## 2.3. Open-LUCIS framework

The LUCIS framework, originally developed for the ESRI-ArcGIS platform, offers a hierarchical, structured, and partici-patory approach to land use planning, specifically focusing on the identification, understanding, and management of land use conflicts [22]. Carr and Zwick provide five steps in using the LUCIS framework: (i) Define Goals, Objectives, and Sub-Objectives, (ii) Data Inventory, (iii) Suitability, (iv) Preference, and (v) Conflict Identification.

LUCIS is ideal for studies of this kind, as HWC at its core is a land use conflict, where both human populations and wildlife compete for resources and space to meet their needs [16,29,30]. As a suitability-based framework, LUCIS assesses land suitability for specific land uses by defining criteria (objectives) that evaluate conditions optimal for each use. Conflict arises when land is deemed suitable for multiple uses, creating competition [31]. Suitability analysis then sets the foundation to identify potential conflict areas, using overlap analysis to determine where suitable zones for different land uses intersect. We choose the LUCIS framework for its goal-oriented, participatory, and forward-looking approach, which is grounded in land-use planning and supports proactive strategies for addressing land-use conflicts [22]. It is grounded in ecological realities while integrating socio-economic considerations. Balancing economic development and environmental conservation remains a pressing issue worldwide, and LUCIS addresses this by providing a structured,

**Table 1. Dataset summary for suitability analysis.**

| Data | Source/Reference | Spatial Resolution | Year | Suitability Analysis Usage |
|------|-----------------|---------------------|------|----------------------------|
| Soil pH, Root Zone Depth & Soil Drainage | ISRIC [1,2] | 250m | 2015 | Agriculture |
| Digital Elevation Model (Slope) | ESA-Copernicus Mission [3] | 30m | 2019 | Agriculture |
| Cattle Post | WildCAT | 30m | 2019 | Agriculture |
| Crop Farms | WildCAT | – | 2023 | Agriculture |
| Roads | OpenStreetMap [4] | – | 2020 | Agriculture; Human Settlement |
| Land Cover | Peace Parks Foundation | 10m | 2021 | Agriculture; Human Settlement |
| Population Density & Distance to Settlement | WorldPop [5,6] | 1km | 2022 | Human Settlement |
| Settlement Layout | WildCAT & Department of Physical Planning, Housing & Estate Management [7] | – | 2023 | Human Settlement |
| Boreholes | WildCAT & Botswana Geoscience Institute | | 2023 | Agriculture; Human Settlement; Wildlife |
| Wildlife Habitat Suitability | WildCRU [8] | 250m | 2023 | Wildlife |
| Land Use Land Cover (LULC) | Author[1] (Modified Land Cover [9] with Existing Land Uses) | 10m | 2024 | Agriculture; Human Settlement |
| Problem Animal Control | Dept. of Wildlife and National Parks, Botswana | – | 2007-2023 | Wildlife |

participatory platform. The framework incorporates land-use priorities through weights assigned by stakeholders and subject matter experts, representing the relative importance of different factors in the analysis [32]. This stakeholder-driven approach enhances their actionability by aligning the results with local priorities in formulating land use plans to deal with and manage land use conflicts. Additionally, the flexibility of LUCIS allows it to integrate a wide array of datasets, from ecological to economic, facilitating a multidimensional approach to conflict analysis.

To address HWCs in Pandamatenga, the study identified agriculture, human settlement, and wildlife (Fig 4) as the primary land-use planning goals. These goals, which mirror the broader objectives of agriculture, urban development, and conservation established in the original LUCIS framework [22], were determined in consultation with WildCAT. The three goals capture the dominant and competing land-use demands in the study area. These goals served as the basis for structuring the LUCIS framework, guiding both the selection of datasets and the development of suitability criteria that informed the overall analysis. These goals were further organized into objectives and sub-objectives, creating a hierarchical structure that defines the specific conditions required to represent each of the competing land-use interests. The suitability layers were refined into preference layers, culminating in the identification of overlapping interests in land (land use conflict). Leveraging the LUCIS framework via the PyLUSATQ (version 0.3.0) plugin [32] in QGIS [33], hence Open-LUCIS, we incorporated the wildlife habitat suitability layer by Loveridge et al. [8] into the study to be analyzed alongside the human-driven goals of agriculture and settlement. Unlike these anthropogenic objectives, wildlife suitability is more ecologically driven and requires specialized expertise to determine, making its integration essential for capturing the ecological dimension of HWC.

**2.3.1. Land suitability analysis.** Land suitability analysis assesses how appropriate a given parcel of land is for a particular purpose, based on a set of criteria to identify optimal spatial patterns for current and future land uses within a region [34–36]. The land units can be cadastral plots, cells of a spatial grid, or a combination of both [37]. Assigning specific land uses to such units involves considering both their physical characteristics and their socio-economic viability [38,39]. In this study, the three major land uses are agriculture, human settlement, and wildlife conservation,

**Fig 4. Conceptual framework for identifying HWC and evaluating the impact of factors.**

corresponding to the goals of the Open-LUCIS model. The Pandamatenga settlement layout, comprising demarcated residential, commercial, and crop farmland vector layers, was merged with the Military Grid Reference System (MGRS) grid vector (68). Using the Merge Vector geoprocessing tool, a composite layer that represented the land units was generated, as shown in Fig 5. A multi-resolution MGRS grid was applied, with 250 m cells near developed areas to capture likely expansion and 1000 m cells in protected zones where less detail was required.

The suitability analysis followed the three-tier hierarchical structure of LUCIS, where sub-objectives informed objectives, which in turn aggregated into overarching goals in a bottom-up manner. At the sub-objective and objective levels, suitability layers were generated with weights between 0 and 1, indicating their relative importance as informed by insights from WildCAT. At each level, we aggregated multiple criteria using a weighted linear combination [40,41] to derive composite layers for the next hierarchy. Additionally, through PyLUSATQ, we employed the Analytical Hierarchy Process [42,43], a widely used Multi-Criteria Evaluation Analysis approach, to aid in the decision-making process [44]. The final suitability composite layers were rescaled from 1 to 9, with 1 indicating the least suitability and 9 the highest suitability [22,24,31]. This standardization process ensured uniformity and consistency across various datasets and criteria, facilitating a more cohesive analysis.

**Agriculture suitability**. The agriculture component of our methodology focuses on two primary goals: crop farming and livestock rearing, which are the predominant agricultural activities in the study area [27]. Given that crop farming is the largest and most dominant agricultural activity in the region, greater weight (0.7) was assigned to crop farming relative to livestock rearing (0.3), which is comparatively less extensive. See Tables 2 and 3 for a detailed breakdown of the weights applied to each sub-goal in the agriculture suitability layer.

For the crop farming goal, primarily representing commercial and smallholder farming, physical suitability was given the highest weight (0.60) to optimize the use of existing farms and associated infrastructure. This included an assessment of soil conditions (root zone depth, soil pH, and drainage levels), terrain characteristics (land cover suitability and slope), and existing development constraints.

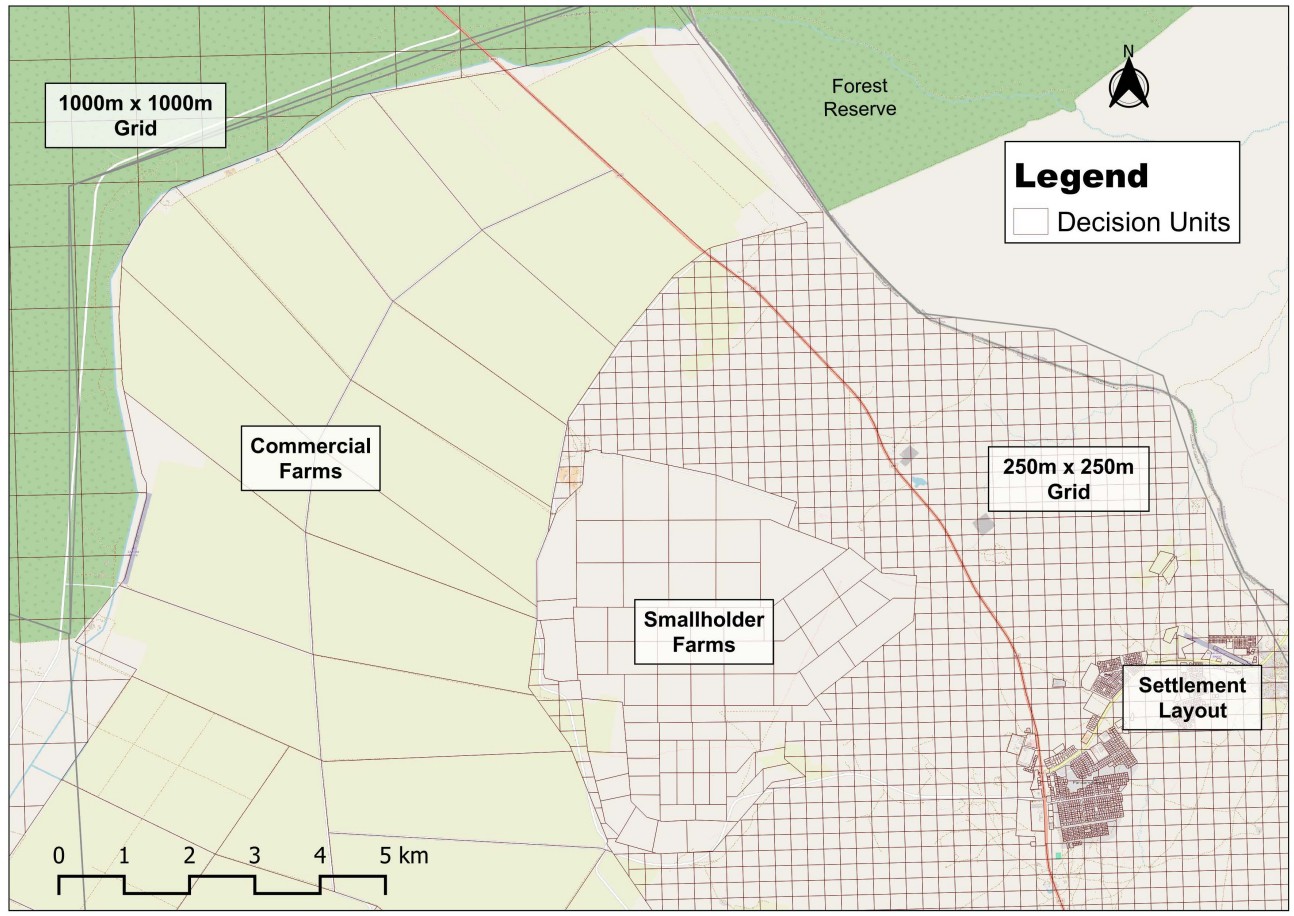

**Fig 5. Map showing the land/decision units used in conducting the study.** Republished from OpenStreetMap Contributors under a CC BY license, with permission from OpenStreetMap Foundation, original copyright 2023.

**Table 2. Livestock suitability parameters and weights.**

| Goal | Objective (Weight) | Objective Weight | Sub-Objective | Sub-Objective Weight |
|------|--------------------|------------------|----------------|----------------------|
| Livestock | Physical | 0.40 | Existing Livestock Location | 0.40 |
| | | | Terrain Characteristics | 0.30 |
| | | | Development Constraints | 0.30 |
| | Economic | 0.30 | Transport Accessibility | 0.33 |
| | | | Water Accessibility | 0.34 |
| | | | Livestock Buffer | 0.33 |
| | Wildlife Condition | 0.30 | Elephant Suitable Habitat | 0.10 |
| | | | Lions Suitable Habitat | 0.50 |
| | | | Leopard Suitable Habitat | 0.20 |
| | | | Spotted Hyaena Suitable Habitat | 0.20 |

**Table 3. Crop farming suitability parameters and weights.**

| Goal | Objective | Objective Weights | Sub-Objective | Sub-Objective Weights |
|------|-----------|-------------------|---------------|-----------------------|
| Crop Farming | Physical | 0.60 | Existing Farms (condition) | 0.30 |
| | | | Soil Condition | 0.20 |
| | | | Terrain Characteristics | 0.20 |
| | | | Development Constraints | 0.30 |
| | Economic | 0.40 | Transport Accessibility | 0.25 |
| | | | Water Proximity | 0.25 |
| | | | Market Accessibility | 0.25 |
| | | | Distance to Crop Farms | 0.25 |

For the livestock goal, the physical objective received slightly greater emphasis (0.40), reflecting the importance of existing livestock locations and infrastructure to encourage continuity in established practices. Terrain characteristics and development constraints were also considered under this category. Economic objectives (0.30) captured accessibility to roads and water sources—especially critical given recurrent drought conditions—alongside compliance with the 300-meter buffer regulation for cattle posts in the Chobe district. Wildlife conditions (0.30) were integrated to minimize exposure to predation, with lions assigned the highest weight (0.50) as they are the most frequently reported and dominant livestock predator in the area, followed by leopards, hyenas, and elephants.

**Human settlement suitability**. This component similarly comprises both physical and economic objectives to ensure the sustainable development of human settlements while minimizing environmental impact and maximizing community well-being. Table 4 shows the weights applied in generating the human settlement suitability layer. To evaluate the physical suitability of potential settlement areas, we considered three key factors, including terrain characteristics, existing land allocations for settlements, and population density, to identify optimal spaces for human habitation within the study area. Terrain characteristics played a significant role in assessing the suitability of potential settlement sites. Factors such as slope and land cover suitability for settlements were evaluated to determine areas suitable for human development. Additionally, land allocations designated for settlements were taken into account to ensure compliance with land use regulations and policies. Population density (persons/km²) served as an indicator of existing settlement patterns and urbanization trends within the study area. By analyzing population distribution, we aimed to concentrate the growth of human settlement to enhance shared existing infrastructure and resources.

Under the economic objective, road accessibility and proximity to already established settlements were considered. Regional and local road networks [45] with varying conditions were assessed to gauge accessibility and connectivity to essential services and amenities. Furthermore, proximity to existing settlements was evaluated to promote contiguous expansion and effective settlement management.

**Table 4. Human settlement suitability parameters with weights.**

| Goal | Objective | Objective Weights | Sub-Objective | Sub-Objective Weights |
|------|-----------|-------------------|---------------|-----------------------|
| Human Development | Physical | 0.50 | Population Density | 0.40 |
| | | | Terrain Characteristics | 0.30 |
| | | | Development Constraints | 0.30 |
| | Economic | 0.50 | Road Accessibility | 0.50 |
| | | | Distance to Settlement | 0.50 |

**Wildlife habitat suitability**. In our wildlife suitability analysis, habitat suitability layers were developed by WildCRU [8] were utilized. Habitat suitability models were developed for larger carnivores and herbivores using spoor and camera trap survey data from across northern Botswana, north-western Zimbabwe, and southern Zambia. The study focused on seven key species: zebra (*Equus quagga*), elephant (*Loxodonta africana*), buffalo (*Syncerus caffer*), lion (*Panthera leo*), leopard (*Panthera pardus*), spotted hyena (*Crocuta crocuta*), and African wild dog (*Lycaon pictus*) that can cause conflicts or are known to travel long distances during dispersal or migration. Data collection was carried out through spoor surveys, segmenting transects into 250 m intervals, with the presence or absence of each species recorded at the centroid of these segments. Additionally, camera trap surveys by WildCRU noted the presence or absence of species at the specific trap locations.

To develop the habitat suitability layers, Loveridge et al. [8] used presence-absence data and analyzed it against a range of environmental (e.g., soil carbon, rainfall, tree cover, land cover) and human-related factors (e.g., proximity to fences, villages, water, roads, population density) through generalized linear mixed models. Each variable was tested at different spatial scales (500 m to 20 km) using univariate models, and the best-fitting scale was selected for further analysis. Variables were standardized (mean-centered and normalized) and filtered to remove highly correlated factors (Pearson correlation threshold of 0.70). A global model for each species was then created using the best predictors from the optimal scales. The Akaike information criterion was applied in a step-by-step process to refine the model. Model performance was validated using the area under the curve (AUC) with 30% of the data reserved for testing. Finally, the habitat suitability maps for different species were combined using the cell statistics tool in QGIS and reclassified with the PyLUSATQ reclassify tool to align with the Open-LUCIS structure, similar to the approach used for human settlement and agriculture suitability.

**2.3.2. Conflict identification based on land use preference.** To simplify the analysis for identifying HWC, the final suitability layers for agriculture, human settlement, and wildlife generated on a 1–9 scale were further binned into a 1–3 scale, denoting low preference, medium preference, and high preference to simplify the conflict identification process [22,31]. A natural breaks (Jenks) classification system achieves categorization by minimizing within-class variance and maximizing between-class variance. This optimization ensures that each class is internally coherent, statistically distinct, and reflective of the natural clustering and distribution of the data, thereby providing class intervals that best capture the distribution [22].

Building on the results of the previously described preference analysis, our study identified areas of potential conflict by assessing overlapping preferences associated with human settlement and/or agriculture (referred to jointly as *human*) against wildlife habitat preferences, thus highlighting HWC. Through overlaying the preference results generated for all three goals, areas demonstrating concurrent suitability for multiple objectives were identified as potential conflict zones, with conflict severity ranging from low to high. The HWC zones that were prone to heightened conflict were defined as areas preferred for agriculture and/or human settlement, overlapping with areas preferred by wildlife. This was operationalized through the following criteria:

*High Conflict: WL=3 And (AG=3 Or HS=3)*

*Moderate Conflict: WL=2 And (AG=2 Or HS=2)*

*Low Conflict: WL=1 And (AG=1 Or HS=1)*

Where WL, AG, and HS represent wildlife, agriculture, and human settlement preferences, respectively. The above processes highlighted are operationalized through the steps shown in Fig 6 below.

## 2.4. Sensitivity analysis

In addition to identifying HWC hotspots in Pandamatenga, we conducted a one-at-a-time (OAT) sensitivity analysis to assess the impact of removing specific map layers or variables on the outcome of the conflict identification. The subjectivity of assigning weights arises from the evaluation of opinions regarding the relative importance of various input variables [44].

Sensitivity analysis helps determine how robust the decision-making process is after ranking the criteria and finding the most influential factors. It also provides further insights into the uncertainty and the relative importance of different factors in driving change, especially when there is more than one set of criteria [46]. To make our results easier to interpret under real-world data uncertainty, we use a map-removal OAT sensitivity analysis [44,46], in which groups of layers representing common land-use criteria are removed at a time, allowing us to isolate the influence of these shared factors with low computational cost and provide a transparent approach for stakeholders. This allowed us to assess how certain environmental and human activities impact the outcome of the conflict. We identified five common factors in the agriculture and human settlement goals (Fig 6), fundamental to the HWC, to assess their influence on the base conflict generated in this study. The factors evaluated include existing land uses, terrain characteristics, road accessibility, water proximity, and development constraints.

## 3. Results and discussion

### 3.1. Land suitability analysis and preference

Land suitability forms the basis for understanding competition among land uses, as each seeks to optimize its objectives within a shared landscape. Within the LUCIS framework, suitability is not an inherent property of the land but a reflection

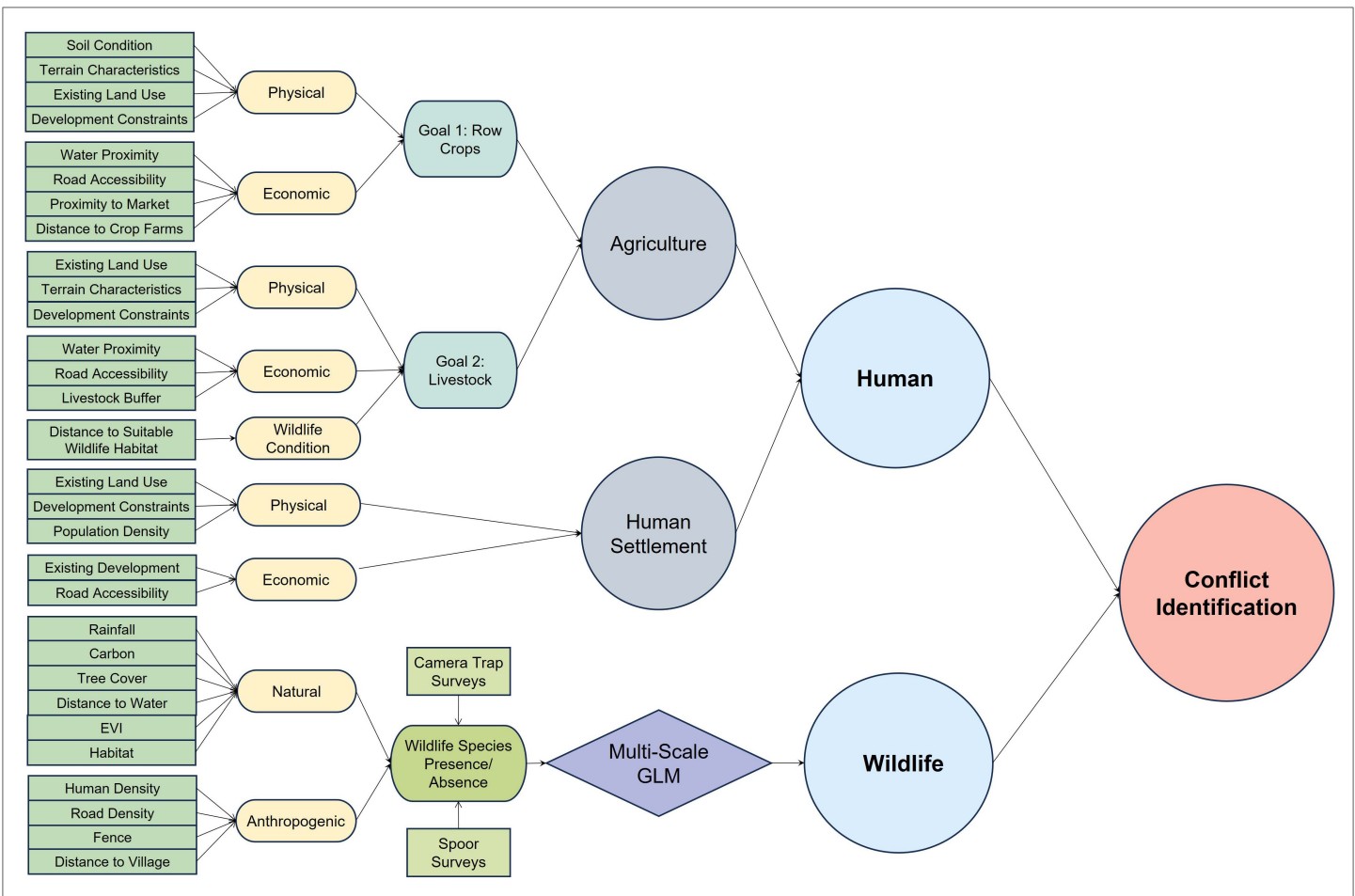

**Fig 6. Geoprocessing flowchart showing the Open-LUCIS framework to identify HWC.**

of model inputs that represent the goals of agriculture, human settlement, and wildlife conservation. This encapsulates the planning process, where competing demands are balanced through spatially explicit objectives. The suitability maps (Fig 7a–7c) illustrate these dynamics. Agricultural suitability, based on a weighted combination of crop farming (70%) and livestock rearing (30%), is concentrated around existing commercial farms and cattle posts. This pattern reflects both the influence of accessibility and existing land uses. Human settlement suitability similarly clustered around pre-existing settlements, where proximity to roads and water services reduces costs and improves convenience. These patterns also reveal the role of development constraints such as land-use regulations, protected areas, and forest reserves in shaping where settlement and agricultural expansion may occur. In contrast, wildlife habitat suitability is more broadly distributed, encompassing large tracts of land that provide forage, water, cover, and space. This pattern is expected, as it reflects the free movement of wildlife across the landscape and aligns with ecological processes that operate independently of human-defined boundaries. In contrast, wildlife habitat suitability is broadly distributed, reflecting expected patterns of free movement and ecological processes that extend beyond human-defined boundaries. The suitability maps were reclassi-fied into three categories—low, moderate, and high (Fig 8a–8c)—to provide a simplified representation of areas associ-ated with agriculture, human settlement, and wildlife habitat. Here, "preference" does not denote literal selection by people or wildlife, but rather the modeled outcome of weights applied to the objectives within the LUCIS framework, offering a basis for identifying potential areas of conflict.

When reclassified into three preference categories (Fig 8a–8c), high-preference areas for agriculture covered approximately 17% of the study area, primarily associated with existing farms and cattle posts. Human settlement preference was minimal, accounting for less than 1% of the area, consistent with the region's small population of 2,728 in 2022 [47]. In contrast, wildlife accounted for about 41% of the high-preference area, reflecting the broad extent of land suitable for conservation.

### 3.2. Human-wildlife conflict identification

In this study, HWC is identified where land units suitable for wildlife habitat overlap with those suitable for agriculture and/or human settlement. This classification should be viewed as a generalized representation of where land-use conflict might occur. It does not distinguish among species-specific interactions (e.g., crop-raiding by elephants versus livestock predation by carnivores) and does not account for temporal dynamics such as seasons and weather. Accordingly, the conflict layers are best interpreted as a demonstration of the framework's application rather than a precise and operational conflict map.

By overlaying the three preference maps, areas of potential land-use conflict were delineated (Fig 9). Approximately 70% of the study area showed no overlap between human and wildlife land-use suitability, largely reflecting zones des-ignated as forest reserves and wildlife concessions. Of the remaining area, 10% was classified as low conflict, 11% as moderate conflict, and 9% as high conflict.

Notably, high-conflict zones were concentrated within and around existing commercial farms. This pattern corresponds with the Chobe District Integrated Land Use Plan, which identifies these farms as being situated along a critical transbound-ary wildlife corridor linking Hwange National Park (Botswana) and Matetsi Safari Area (Zimbabwe) to Chobe National Park [19]. The availability of boreholes, cultivated fields, and cattle posts in these areas further increases the likelihood of conflict by attracting elephants and large carnivores [48]. Although less extensive, low and moderate-conflict areas reflect meaning-ful overlaps between human and wildlife land-use suitability. These zones highlight potential points of contention that merit attention, as they could intensify into more severe conflicts if not addressed through proactive management.

### 3.3. Sensitivity analysis

An OAT sensitivity analysis was conducted by sequentially removing five common factors in all three land uses from the baseline model in Fig 6, including existing land use, terrain characteristics, development constraints, road accessibility,

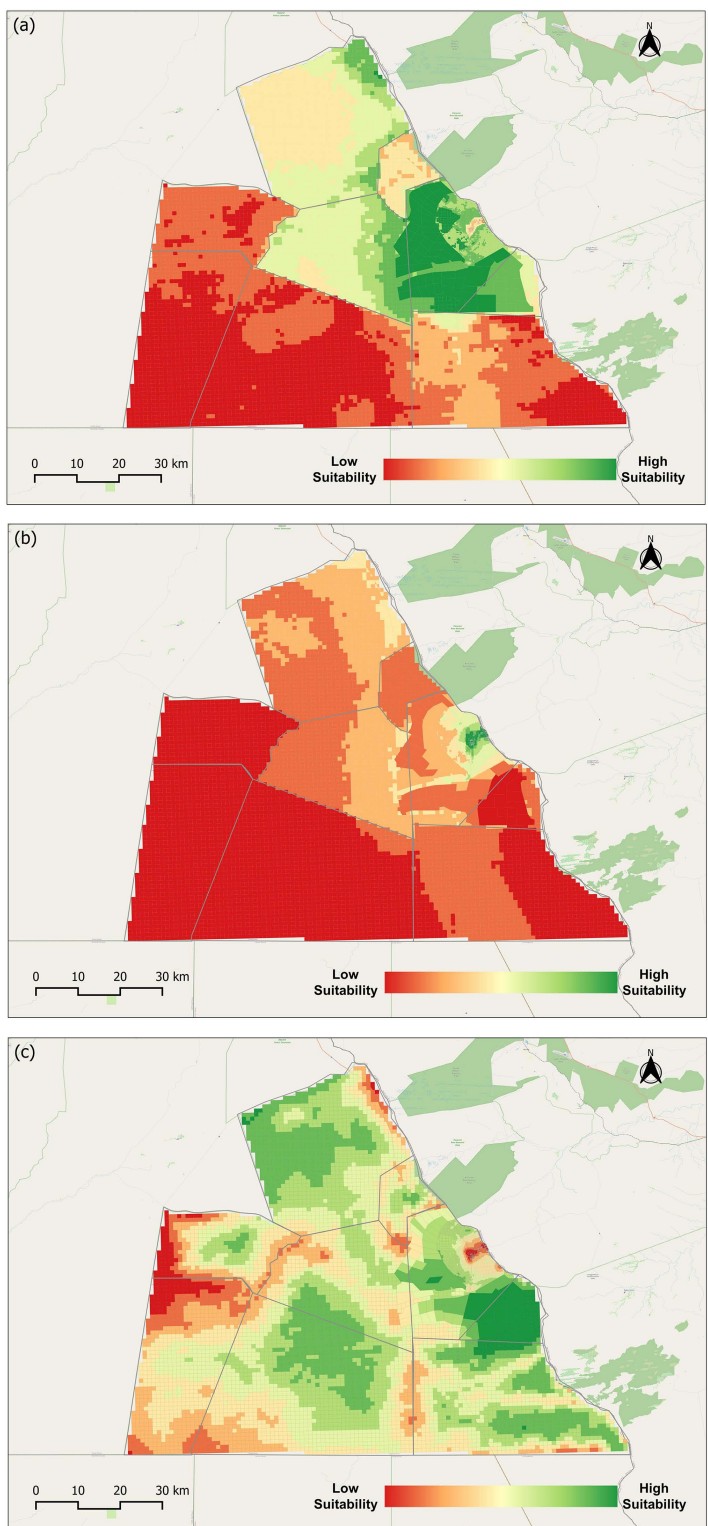

**Fig 7. Suitability maps for Pandamatenga representing goals of each land use.** (a) Agriculture suitability map; (b) Human settlement suitability map; and (c) Wildlife habitat suitability map. Republished from OpenStreetMap Contributors under a CC BY license, with permission from OpenStreet-Map Foundation, original copyright 2023. Foundation, original copyright 2023.

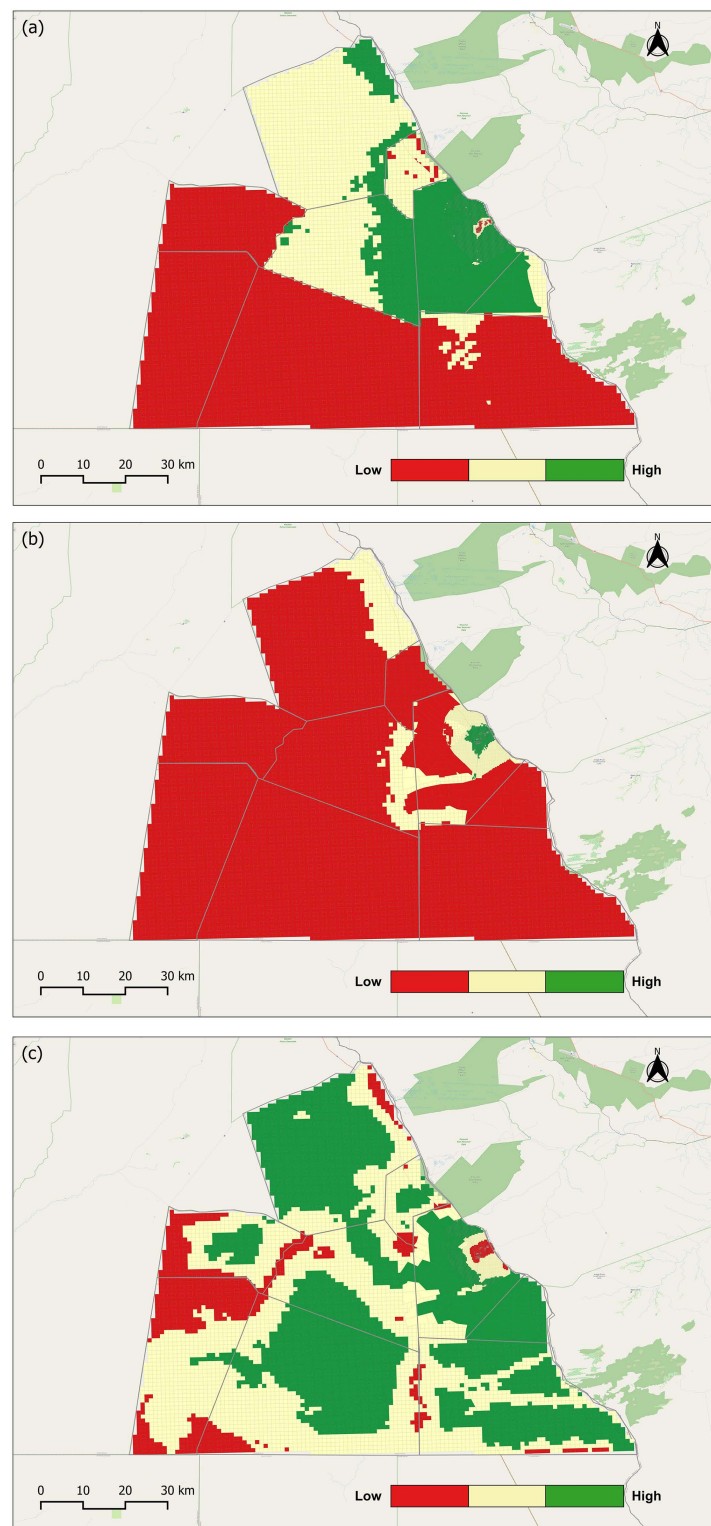

**Fig 8. Preference maps representing reclassification of suitability maps for each land use.** (a) Agriculture land use preference map; (b) Human settlement land use preference map; and (c) Wildlife habitat land use preference map. Republished from OpenStreetMap Contributors under a CC BY license, with permission from OpenStreetMap Foundation, original copyright 2023.

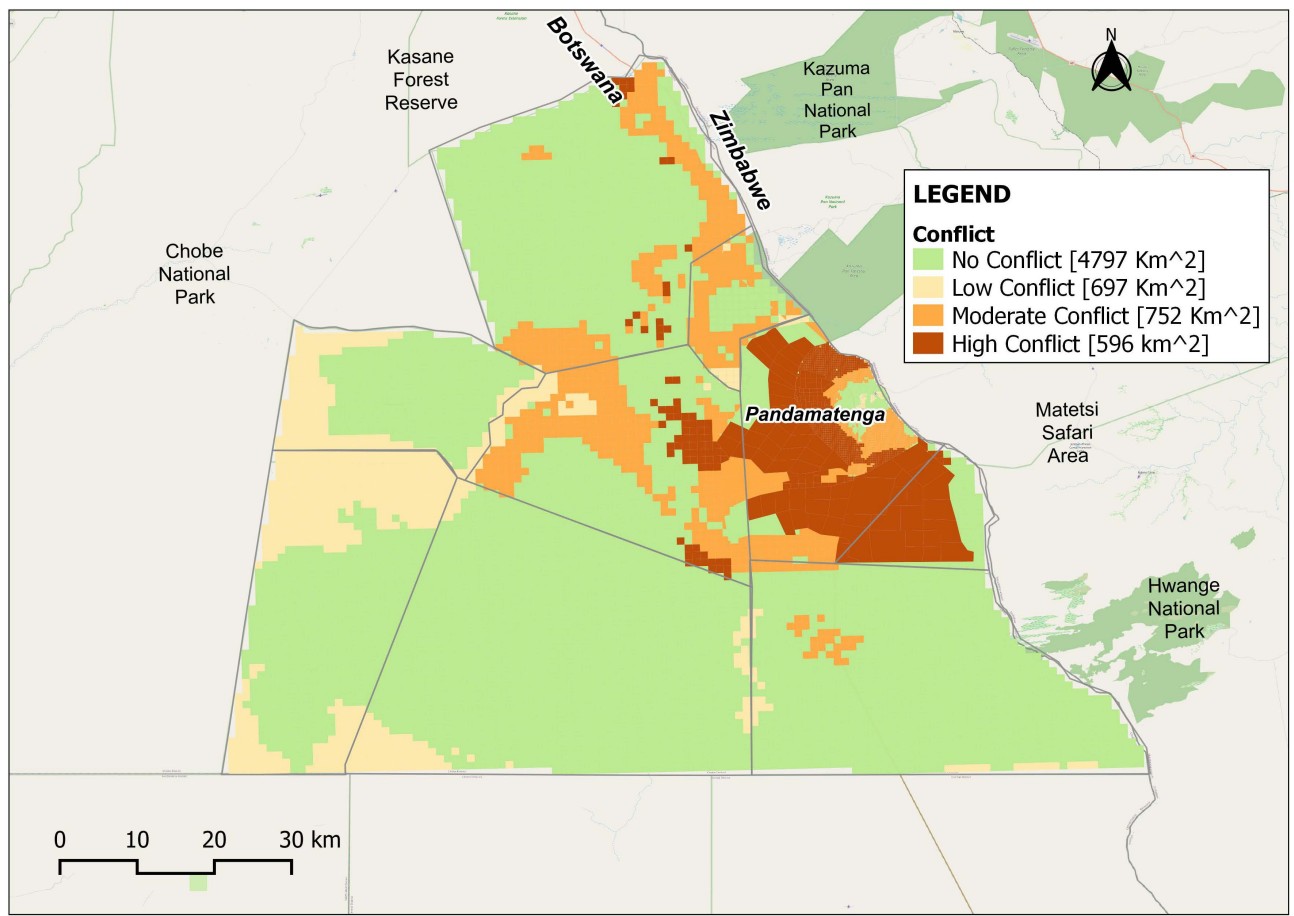

**Fig 9. Base HWC map showing characterization of the conflict in Pandamatenga.** Republished from OpenStreetMap Contributors under a CC BY license, with permission from OpenStreetMap Foundation, original copyright 2023.

and water proximity. As shown in Table 5, conflict levels declined when existing land use, terrain characteristics, and development constraints were removed; however, they increased with the exclusion of road accessibility and water proximity. Among these factors, existing land use, road accessibility, and development constraints had the most significant impact on high-conflict areas.

The table above summarizes the distribution of HWC zones under different scenarios based on the removal of individual factors from the suitability analysis. Relative to the baseline scenario, excluding existing land use and development constraints reduced high-conflict areas by 273 km² (−45.8%) and 213 km² (−35.7%), respectively, highlighting their dominant role in shaping conflict dynamics. The existing land use factor represents current human activities on the landscape, including commercial farms, settlements, and livestock-rearing areas. When this factor is excluded, parcels are treated as undeveloped, effectively reverting to their natural state. A substantial decrease in high conflict is not surprising, as the existing commercial farms—making up the majority of the high-conflict area—have been established in locations critical to wildlife movement or habitation, creating points of intense conflict, as corroborated by [49].

Removing terrain characteristics, which capture land cover suitability and slope, produced only a negligible reduction of 15 km² (−2.5%) in high-conflict areas compared to the base scenario, suggesting limited influence at the scale of this analysis. In contrast, excluding development constraints, which define permissible zones for farming, settlement, and

**Table 5. Map removal sensitivity analysis metrics showing changes in conflict dynamics (Numbers in red show significantly high conflict changes).**

| Removed Factors | Intensity | Area (km²) | Change (km²) | Percentage Change |
|---|---|---|---|---|
| Base Conflict (Non-Removal) | No | 4,798 | N/A | N/A |
| | Low | 697 | N/A | N/A |
| | Moderate | 752 | N/A | N/A |
| | High | 596 | N/A | N/A |
| Existing Land Use | No | 4,933 | 135 | 2.8% |
| | Low | 692 | −5 | −0.7% |
| | Moderate | 895 | 143 | 19.0% |
| | High | 323 | **−273** | **−45.8%** |
| Terrain Characteristics | No | 4,632 | −166 | −3.5% |
| | Low | 693 | −4 | −0.57% |
| | Moderate | 937 | 185 | 24.6% |
| | High | 581 | −15 | −2.5% |
| Road Accessibility | No | 4,690 | −107 | −2.2% |
| | Low | 699 | 2 | 0.3% |
| | Moderate | 697 | −55 | −7.3% |
| | High | 756 | **160** | **26.8%** |
| Water Proximity | No | 4,578 | −220 | −4.6% |
| | Low | 694 | −3 | −0.4% |
| | Moderate | 929 | 177 | 23.5% |
| | High | 642 | 46 | 7.7% |
| Development Constraints | No | 5,090 | 292 | 6.1% |
| | Low | 746 | 49 | 7.0% |
| | Moderate | 624 | −128 | −17.0% |
| | High | 383 | **−213** | **−35.7%** |

livestock rearing, reduced high-conflict areas by 213 km² (−35.7%). While such constraints are intended to regulate land use, if not strategically undertaken, they can also concentrate human activities within ecologically sensitive areas, thereby intensifying overlap with wildlife. In their absence, land uses are more flexibly distributed across the landscape, reducing direct competition and lowering conflict potential.

The marginal increase in high-conflict areas following the exclusion of water proximity by 46 km² (+7.7% from the base scenario) highlights the importance of water sources as focal points for both human and wildlife activities. Designated boreholes for domestic use, livestock, and agriculture help concentrate interactions within specific zones, thereby limiting the wider spread of conflict. Without this structuring effect, competition for water becomes more spatially diffused, increasing the likelihood of conflict across the landscape. Similarly, the removal of road accessibility expanded high-conflict areas by 160 km² (+26.8%), reflecting the role of transport corridors in concentrating human activity. In their absence, development disperses more widely, potentially leading to greater encroachment into wildlife habitats and heightened conflict risk.

## 4. Conclusion

This study demonstrates how the Open-LUCIS framework can be utilized in assessing HWC dynamics. By integrating a variety of geospatial data with domain and expert knowledge from WildCAT, we demonstrated how agriculture, human settlement, and wildlife habitat suitability interplayed to reveal areas of low, medium, and high conflict. The analysis, in particular, highlighted a high level of land use conflict where commercial farms overlap with transboundary wildlife corridors.

Sensitivity analysis further showed that existing land use, development constraints, and road accessibility strongly influence conflict dynamics, drawing focus to the critical role of planning decisions in shaping outcomes.

The implications extend beyond Pandamatenga. By operationalizing Open-LUCIS, this study demonstrates that resource-limited regions, in terms of GIS data and software, can conduct rigorous and transparent assessments. Such accessibility, when paired with locally grounded expertise, symbolizes a shift toward more equitable and participatory approaches to land-use planning. The conflicts observed in Chobe are not isolated but reflect broader global patterns in which rapid development intersects with ecological imperatives. In this wider context, Open-LUCIS provides a transferable pathway for navigating these tensions, offering planners and policymakers in developing countries a cost-effective tool to balance development objectives with the need to safeguard ecological integrity in sensitive areas worldwide.

Nevertheless, this study highlights areas that warrant deeper investigation. Seasonal and climatic variability, such as rainfall fluctuations and drought, was not considered, though these factors shape HWC dynamics. Due to data availability in rural Botswana areas, the low-resolution data may have obscured local patterns, and the sensitivity analysis, while transparent, was not validated against empirical conflict records. The framework's computational demands may also restrict use in local offices. Future research should address these gaps to improve adaptability and predictive capacity for understanding the complex dynamics of HWC.

## Acknowledgments

The authors would like to thank the Chobe Land Board, the Department of Physical Planning under the Chobe District Council, for providing data support, the Peace Parks Foundation, and the Department of Wildlife and National Parks, Botswana. Also, special thanks to the WildCAT Botswana Trust, Maun, Botswana, for providing data, expert knowledge, and local insight.

## Author contributions

**Conceptualization:** Silas Achidago, Changjie Chen, Jasmeet Judge.

**Data curation:** Silas Achidago, Mogae Makonyela, Lynn Fanikiso, Lara Sousa, Robynne Kotze, Andrew Loveridge.

**Formal analysis:** Silas Achidago, Changjie Chen.

**Funding acquisition:** Jasmeet Judge, Jess Isden.

**Investigation:** Silas Achidago, Mogae Makonyela, Lynn Fanikiso, Lara Sousa, Robynne Kotze, Andrew Loveridge.

**Methodology:** Silas Achidago, Changjie Chen, Jasmeet Judge.

**Project administration:** Jasmeet Judge.

**Resources:** Changjie Chen, Jasmeet Judge, Gregory Kiker, Jess Isden.

**Software:** Silas Achidago, Changjie Chen.

**Supervision:** Changjie Chen, Jasmeet Judge, Gregory Kiker.

**Visualization:** Silas Achidago, Changjie Chen, Jasmeet Judge.

**Writing – original draft:** Silas Achidago.

**Writing – review & editing:** Silas Achidago, Changjie Chen, Jasmeet Judge, Lara Sousa, Robynne Kotze, Gregory Kiker, Kedisaletse Selume, Kim Young, Robin Lines, Jess Isden, Andrew Loveridge, Yan Wang, Aditya Singh.

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
