## [Decision Letter · Decision Letter 0]

28 Jul 2025

Thank you for submitting your manuscript to PLOS ONE. After careful consideration, we feel that it has merit but does not fully meet PLOS ONE’s publication criteria as it currently stands. Therefore, we invite you to submit a revised version of the manuscript that addresses the points raised during the review process.

The manuscript is a useful contribution by way of application of an open-source suitability-based framework for modeling human-wildlife conflicts in Botswana, which is highly appropriate to PLOS ONE's focus on methodological soundness and data-informed research. However, both reviewers also point out several issues that must be revised. Some of the key enhancements include providing explicit details on stakeholder consultations, data collection protocols, and the basis of weight assignments, with appropriate uncertainty estimation or sensitivity analyses. The introduction and discussion should be shortened to focus on the study's specific aims, avoiding general or subjective statements. Descriptions of data preprocessing, model validation and classification techniques are also absolutely required. These changes, once addressed, will significantly enhance the manuscript's clarity, reproducibility, and scientific impact.

We look forward to receiving your revised manuscript.

Kind regards,

Dereje Yazezew Mammo, Ph.D.

Academic Editor

PLOS ONE

Journal Requirements:

4. We note that Figures 1,2,3,5,7a,7b,7c,8a,8b,8c and 9 in your submission contain [map/satellite] images which may be copyrighted. All PLOS content is published under the Creative Commons Attribution License (CC BY 4.0), which means that the manuscript, images, and Supporting Information files will be freely available online, and any third party is permitted to access, download, copy, distribute, and use these materials in any way, even commercially, with proper attribution. For these reasons, we cannot publish previously copyrighted maps or satellite images created using proprietary data, such as Google software (Google Maps, Street View, and Earth). For more information, see our copyright guidelines: http://journals.plos.org/plosone/s/licenses-and-copyright.

1. You may seek permission from the original copyright holder of Figures 1,2,3,5,7a,7b,7c,8a,8b,8c and 9 to publish the content specifically under the CC BY 4.0 license.

Reviewers' comments:

Reviewer's Responses to Questions

**Comments to the Author**

1. Is the manuscript technically sound, and do the data support the conclusions?

Reviewer #1: Yes

Reviewer #2: Yes

2. Has the statistical analysis been performed appropriately and rigorously?

Reviewer #1: Yes

Reviewer #2: I Don't Know

3. Have the authors made all data underlying the findings in their manuscript fully available?

Reviewer #1: Yes

Reviewer #2: No

4. Is the manuscript presented in an intelligible fashion and written in standard English?

Reviewer #1: Yes

Reviewer #2: Yes

Reviewer #1: Title: Identifying and Characterizing Human-Wildlife Conflicts Using an Open-Source Suitability-Based Framework: A Case Study in Botswana.

1. Review of the Introduction Section

Strengths

o The introduction effectively sets the stage by highlighting the global scale of HWC(e.g., Kenya, India, China) and its socio-economic impacts (crop loss, fatalities).

o Strong emphasis on policy responses (e.g., KAZA Transfrontier Conservation Area, UNDP initiatives) to demonstrate real-world relevance.

o Critiques shortcomings of current mitigation strategies (e.g., electric fencing, tech-based alerts) and argues for land-use planning (LUP) as a systemic solution.

o Clearly positions Open-LUCIS as an improvement over proprietary LUCIS by being open-source, participatory, and reproducible.

o Introduces sensitivity analysis as an innovative addition to assess factor influence on HWC.

o Progresses smoothly from global → national → local perspectives, culminating in the study’s specific objectives.

Weaknesses and Concerns

o While global cases (Kenya, India) are useful, the link to Botswana’s Chobe District (study area) is underdeveloped. A paragraph contrasting global trends with local Chobe dynamics (e.g., elephant conflicts, agricultural expansion) would strengthen focus.

o Claims LUCIS is "effective" but cites UNDP procurement notices (41,42) as evidence rather than peer-reviewed outcomes. Needs specific examples of LUCIS reducing HWC (e.g., success metrics from Botswana’s Tawana/Chobe).

o Open-LUCIS’s limitations (e.g., computational demands, stakeholder bias in weighting) are not addressed.

o Lacks discussion of competing land-use frameworks (e.g., Marxan, InVEST) to justify why LUCIS is superior for HWC.

o No mention of spatial ecology theories (e.g., landscape permeability, source-sink dynamics) that ground habitat-suitability modeling.

o Mentions "domain knowledge" and "local priorities" but doesn’t detail how stakeholders were involved (e.g., workshops, surveys) or how their inputs were weighted.

2. Review of the Material and methods Section

Strengths

o The use of the Open-LUCIS framework (adapted from LUCIS) is well-justified, given its participatory, hierarchical, and suitability-based approach to land-use conflict analysis.

o Clear articulation of the three primary land-use goals (agriculture, human settlement, wildlife conservation) and their sub-objectives ensures methodological transparency.

o The study leverages diverse spatial datasets (e.g., soil properties, DEM, land cover, wildlife habitat suitability) from reputable sources (ISRIC, ESA-Copernicus, WildCRU, etc.), enhancing reproducibility.

o Multi-resolution MGRS grid (250 m and 1000 m) is a pragmatic choice to balance detail and computational efficiency. But I have some concern which is given in the weaknesses and concerns section.

o Weighted Linear Combination (WLC) and Analytical Hierarchy Process (AHP) are appropriately applied for suitability analysis, aligning with established multi-criteria decision-making (MCDM) practices.

o Sensitivity analysis (map removal approach) is a strong addition to assess the robustness of the model, particularly for dynamic human-driven factors.

o The 1–9 to 1–3 reclassification (using natural breaks) and conflict criteria (high/moderate/low) are logically defined and operationalized.

Weaknesses and Concerns

o While the study mentions stakeholders, there is no information on:

How their inputs were collected (workshops, surveys?) and integrated into the weighting process.

Potential biases in stakeholder selection (e.g., overrepresentation of certain groups).

o The weights for suitability analysis (Tables 2–4) are attributed to WildCAT Botswana Trust and domain experts, but:

No justification is provided for why specific weights were chosen (e.g., why "Physical" = 0.40 for livestock).

Lack of uncertainty quantification (e.g., confidence intervals for weights). Uncertainty Quantification for Weight Justification in Suitability Analysis can be analyzed through expert elicitation with confidence intervals or probabilistic sensitivity analysis or global sensitivity analysis.

o The wildlife habitat suitability layer (from WildCRU) relies on presence-absence data and GLMMs, but:

Temporal mismatch: Data years are unclear (e.g., spoor surveys). Are they representative of current conditions?

Scale dependency: The 250 m resolution may not capture fine-scale wildlife movements (e.g., elephants). In my opinion, A 250 m resolution grid may not accurately capture elephant habitat needs because it averages large areas into single cells. Elephants use fine-scale features (like 100 m-wide river corridors or small forest patches), but these critical details get "blurred" when combined with surrounding unsuitable land in a 250 m cell. This can lead to errors—either overlooking important habitats (false negatives) or mislabeling fragmented areas as suitable (false positives). Essentially, the model might miss where elephants actually go or overestimate safe zones. I suggest authors to address this.

o The OAT (one-at-a-time) approach is limited; a global sensitivity analysis (e.g., Monte Carlo) would better account for interactions between variables.

o No discussion of how much change in output (e.g., % conflict area shift) was observed after factor removal.

o PyLUSATQ plugin: How was it validated for this study? Is it open-source?

o Land unit generation: The process of creating the composite layer (Figure 5) needs more clarity (e.g., how were MGRS grids merged with settlement layouts?).

o Data preprocessing: Were datasets harmonized to common resolutions/projections?

o The high/moderate/low conflict classification (WL=3 & AG=3, etc.) may oversimplify:

No consideration of species-specific conflicts (e.g., elephants vs. lions).

No temporal dynamics (e.g., seasonal migration impacts).

3. Review of Results & Discussion Section

Strengths

o Figures 7a-c and 8a-c effectively visualize land-use preferences, with quantifiable results (e.g., 17% high-agriculture suitability, 41% high-wildlife preference).

o Logical progression from suitability → preference → conflict identification.

o Transparent criteria for conflict classification (low/moderate/high) based on overlapping preferences.

o High-conflict zones align with known wildlife corridors (e.g., commercial farms near Hwange-Chobe corridor), validating the model.

o Table 5 briefly summarizes factor impacts, highlighting trade-offs between conservation and development.

o Emphasizes adaptive land-use planning (e.g., revising zoning maps, habitat restoration) and stakeholder engagement.

Weaknesses & Concerns

o No discussion of model validation (e.g., comparison with historical HWC incident data). Are high-conflict zones empirically verified?

o Weighting justification: Crop farming (70%) vs. livestock (30%) weights lack empirical or stakeholder-derived rationale.

o The 250 m resolution (noted in Methods) may misclassify fine-scale conflicts (e.g., elephant crop-raiding at farm edges). Not addressed in Results. Authors have to address this.

o OAT limitations: Ignores factor interactions (e.g., roads + water proximity may synergistically increase conflict). A global sensitivity method would strengthen findings.

o Missing climate variables: Droughts or rainfall shifts could alter water-driven conflicts but are excluded.

o Stakeholder input is mentioned but not quantified (e.g., how local knowledge refined weights).

Final Recommendation

The methodology is largely sound but requires greater transparency in stakeholder engagement, weight justification, and sensitivity analysis to meet peer-review standards. Authors must address or provide the justifications for the “Weaknesses and concerns” of Introduction, materials and methods as well as results and discussion sections.

Decision: Major Revisions Required.

Reviewer #2: Review of HWC Botswana

This is an interesting, informative, if lengthy manuscript that reports on a study using a “suitability framework” and the open-source Land Use Conflict Identification Strategy tool.

The introduction is lengthy and reads more like a general review of human-wildlife conflict (HWC) and other topics rather than a concise introduction to their study. As a result, the lead-up to the description of study goals is less clear. The authors include details and examples from numerous studies that are minimally relevant to this study. The authors also insert numerous subjective statements about what should be done (e.g., line 127: “To effectively manage HWC, an integrated…approach…is required.” ). It would be helpful to describe the spatial extent and temporal scale in the goals section. The introduction of the LUCIS framework also comes out of nowhere (and it is not entirely clear at first this is for the Chobe District). The paragraph describing generically what is in the methods section and results can be deleted.

The methods are generally informative, but some details could be described in greater detail (e.g., not all readers will know about “the” WildCRU (e.g., simply explain its relationship to Oxford), and it is not clear what the “domain knowledge” from WildCRU entails. The study area description probably includes more detail than is needed for this study.

How or why did the study identify agriculture, human settlement, and wildlife as the primary LUP goals (line 260)? On page 13 the authors state that the weights were based on “domain knowledge and expert opinion” from the WildCAT Botswana Trust, but the authors do not explain how this was derived and cite an unpublished report as the source.

The results are interesting but largely a function of the weights applied in the model. Habitat suitability models in general suffer from a risk of subjective weighting. For example, the authors describe and “objective weight” (table 2) of 0.4 for physical, 0.3 economic, and 0.3 for wildlife for livestock (and do this again for crop farming and human development). These were further subdivided by “sub-objective weights”. The results of the model would be very different if different weights were selected, and there was no effort to provide a range of weights, or variances, that could provide a range of possible outcomes.

The results also include some management goals that seem at odds with research questions (e.g., line 341: “…we aimed to concentrate the growth of human settlement to enhance shared existing infrastructure and resources.” Why? This seems like an important a priori decision).

The authors use a Jenks classification to categorize the suitability layers. This classification is typically used to maximize color separation on maps—but it can lead to extremely varied classes to achieve that goal. For example, one class could include an extremely large range while another class could include an extremely small range, depending on the histogram. How is this related to “preference”? The authors may want to further clarify why this provides (line 381) “…optimal and significant class intervals.”

The manuscript is generally well written and edited. However, the manuscript includes some very lengthy paragraphs (e.g., paragraph starting p 5 line 113 and ending page 7 line 148!) that are difficult to read and could be shortened into focused and concise paragraphs.

The authors carried out a sensitivity analysis but do not include validation steps to help us understand whether the weights they selected are relevant in the real world. Understanding which variables impact the model more or less (the sensitivity) is helpful, but without some kind of validation steps it is just a hypothetical model. What would happen if different weights were used? At the very least, the authors should discuss the limitations of this type of subjective HSI approach.

The authors provide a series of specific “preferences” (e.g., no substantial overlap in land use preferences between people and animals in 70% of the examined region…”), but how much of this is simply a function of separating conserved wildlands from human settlements and agricultural/livestock grazing lands? What does it mean that “41% of the region…is preferred by wildlife.”? Does wildlife really “prefer” just some of this land, or is this simply a function of their model parameters?

The discussion, like the introduction is lengthy and includes statements that are unnecessary (e.g., line 529: “This study fundamentally addresses the emergence of HWC….” Really? Or line 531: “The results of this study are particularly crucial for land administrators…” says who?). The authors would benefit from a narrower and more focused discussion of their results, the strengths and limitations of their methods, future directions, and the narrower set of management implications that deriver from their results rather than repeating statements about how crucial their results are.

The conclusion section is unnecessary and is largely repetitive. Anything novel should be included in the discussion.

I would encourage the authors to: shorten and focus the introduction and discussion, remove subjective “should statements” from the introduction and provide concise recommendations relevant to their study results in the discussion, the description of how the weights were developed needs more clarification in the methods section, and the discussion would benefit from a description of possible limitations (including why there was no validation step). The title of the paper might be more informative if it was something like “applying the LUCIS framework to HWC in Botswana” or something similar.

With some additional editing, I am confident this will be a nice contribution to our understanding of human-wildlife conflict and land use conflicts in Botswana.

**Do you want your identity to be public for this peer review?** For information about this choice, including consent withdrawal, please see our Privacy Policy

Reviewer #1: No

Reviewer #2: No

---

## [Author Response · Author response to Decision Letter 1]

30 Sep 2025

We thank the editor so much for the constructive feedback and for recognizing the contribution of our manuscript. We carefully considered all comments and have revised the manuscript accordingly.

We expanded the methodology to provide explicit information on stakeholder consultations, clarifying how domain knowledge and expert input informed weight assignments and clearly stating the limitations. We also added a subsection describing data sources, collection protocols, and quality assurance measures to improve transparency. The basis of weight assignments was explained more clearly, supported by literature and expert rationale, and the section on uncertainty estimation was strengthened by elaborating on the sensitivity analysis used. This highlighted how robustness and decision-relevance were assessed under data uncertainty. The introduction and discussion were streamlined to focus on the specific aims, results, limitations, and implications of the study while avoiding general or subjective statements. This sharpened the narrative and improved readability.

We are confident that the revisions made are adequate and fully address all concerns raised by the editor and reviewers, and that the manuscript now meets the standards of clarity and methodological rigor required by PLOS One.

---

## [Decision Letter · Decision Letter 1]

23 Oct 2025

Applying the Open-LUCIS Framework to Identify and Characterize Human–Wildlife Conflicts:  A Case Study in Botswana

PONE-D-25-30977R1

Dear Dr. Silas Achidago,

Based on the reviewers' advice, and my own reading of the manuscript, we’re pleased to inform you that your manuscript has been judged scientifically suitable for publication and will be formally accepted for publication once it meets all outstanding technical requirements.

Kind regards,

Dereje Yazezew Mammo, Ph.D.

Academic Editor

PLOS ONE

---

## [Editor Report · Acceptance letter]

PONE-D-25-30977R1

PLOS One

Dear Dr. Achidago,

I'm pleased to inform you that your manuscript has been deemed suitable for publication in PLOS One. Congratulations! Your manuscript is now being handed over to our production team.

Kind regards,

on behalf of

Dr. Dereje Yazezew Mammo

Academic Editor

PLOS One